# Progress and Challenges in the Use of a Liver-on-a-Chip for Hepatotropic Infectious Diseases

**DOI:** 10.3390/mi12070842

**Published:** 2021-07-19

**Authors:** Kasem Kulkeaw, Worakamol Pengsart

**Affiliations:** 1Department of Parasitology, Faculty of Medicine Siriraj Hospital, Mahidol University, Bangkok 10700, Thailand; 2Faculty of Graduate Studies, Mahidol University, Nakhon Pathom 73170, Thailand; bbananx@gmail.com

**Keywords:** liver-on-a-chip, hepatocytes, induced pluripotent stem cell, infectious diseases, hepatitis virus, malaria

## Abstract

The liver is a target organ of life-threatening pathogens and prominently contributes to the variation in drug responses and drug-induced liver injury among patients. Currently available drugs significantly decrease the morbidity and mortality of liver-dwelling pathogens worldwide; however, emerging clinical evidence reveals the importance of host factors in the design of safe and effective therapies for individuals, known as personalized medicine. Given the primary adherence of cells in conventional two-dimensional culture, the use of these one-size-fit-to-all models in preclinical drug development can lead to substantial failures in assessing therapeutic safety and efficacy. Advances in stem cell biology, bioengineering and material sciences allow us to develop a more physiologically relevant model that is capable of recapitulating the human liver. This report reviews the current use of liver-on-a-chip models of hepatotropic infectious diseases in the context of precision medicine including hepatitis virus and malaria parasites, assesses patient-specific responses to antiviral drugs, and designs personalized therapeutic treatments to address the need for a personalized liver-like model. Second, most organs-on-chips lack a monitoring system for cell functions in real time; thus, the review discusses recent advances and challenges in combining liver-on-a-chip technology with biosensors for assessing hepatocyte viability and functions. Prospectively, the biosensor-integrated liver-on-a-chip device would provide novel biological insights that could accelerate the development of novel therapeutic compounds.

## 1. Introduction

Individualized genetic factors contribute to disease susceptibility and severity [1,2], drug tolerance [3], and drug-induced liver injury (DILI) [4]. Some genetic loci are potentially translatable to clinical use, for example, the prediction of treatment responsiveness in hepatitis C infection [5], severe adverse events caused by the viral replication inhibitor abacavir in HIV treatment [6], and precision therapy in candidiasis [7]. Prospectively, treatment regimens must meet the medical needs of individuals for higher effectiveness and safety. Thus, personalized medicine has emerged as a form of medical care (diagnosis, treatment and prevention) designed according to individual genetic characteristics relevant to disease susceptibility, disease progression, and responsiveness to specific treatments [8,9].

Knowledge in stem cell biology, bioengineering and material sciences has advanced In Vitro models capable of recapitulating human tissue and organs. Examples include organoids and organ-on-chip assays. Organoids are three-dimensional (3D) structures of cells derived from stem cells, including pluripotent stem cells or adult stem cells. The organoid generally grows in a semisolid medium and consists of different cell types that self-organize in a hierarchical manner similar to that of a tissue [10]. Thus, organoids represent physiologically relevant systems for dissecting tissue development [11,12] and modeling human diseases [13], including infectious diseases [14,15]. Applications of organoids are comprehensively reviewed by Kim et al. [16]. In contrast, an organ-on-chip is cells layered on a microscale plastic plate (called a chip) with a flow of fluid in a tiny channel. It provides a highly dynamic microenvironment relative to the static nature of the organoid. Thus, greater physiological relevance to human organs could be obtained, including the liver [17,18], lung [19], kidney [20], heart [21], intestine [22] or multiple organs [23]. Both systems are capable of functioning as human tissue In Vivo at efficiencies higher than those of conventional 2D cell culture. Thus, organoids and organs-on-a-chip have emerged as powerful tools to personalize treatment and assessments of drug safety [8,9]. Moreover, in the pipeline of drug development, the preclinical phase is important for assessing drug efficacy and safety prior to entering clinical trials. Culture of cancer-derived or immortalized cell lines and animals is the mainstay for assessing drug toxicity. Given its non-physiological relevance to the human body, this approach often leads to the failure of many candidate chemical compounds or drugs in clinical trials [24,25] and post-marketing withdrawal [26]. Regarding the increasing recognition and realization of personalized treatment, these conventional, one-size-fit-to-all methods have been revisited, leading to the use of tailored organoids and organs-on-a-chip.

The liver is a target organ of pathogenic microorganisms, including hepatitis viruses and *Plasmodium* spp., the causative agents of hepatitis and malaria, respectively. Although drugs are available for these devastating infections, treatment responsiveness, drug resistance and, to a lesser extent, DILI reduce therapeutic efficacy. Therefore, more effective and safer drugs are needed. Here, we review evidence of individual responses to drugs against hepatotropic infectious diseases and advances in current liver-on-a-chip models of those diseases and drug-induced toxicity. Regarding the variation in drug response among individuals, methods to design personalized liver-on-a-chip systems are discussed with a focus on the source of patient-specific cells. Moreover, a combination of liver-on-a-chip technology with biosensors is reviewed for the assessment of hepatocyte viability and functions, and some novel biological insights are needed for the development of therapy.

## 2. Necessity of Personalized Medicine for Hepatotropic Infectious Diseases

Hepatocytes are targets of pathogenic microorganisms such as hepatitis virus and *Plasmodium* spp., which are causative agents of hepatitis and malaria, respectively. Viral hepatitis and malaria remain global life-threatening infectious diseases. In 2019, over 3 million people were newly infected with chronic hepatitis B and C virus [27]. Viral hepatitis caused 1.34 million deaths in 2015, a higher number than HIV-related deaths. The major causes of death were cirrhosis and primary liver cancer [28]. The WHO has observed an increase in viral hepatitis-induced death since 2000 [27]. Relative to other hepatitis viruses, more than 90% of all hepatitis virus-caused mortality is due to hepatitis B and C-related complications such as cirrhosis and hepatocellular carcinoma [27]. For malaria, the World Health Organization reported that the numbers of malaria cases and deaths have not substantially changed since 2010. Between 2010–2019, more than 200 million cases of malaria and more than 400,000 deaths were reported worldwide [29]. Advances in pharmacogenomics, the identification of genetic factors associated with variability in drug responses, including metabolism, and the extent of drug-induced liver injury have emerged. Because hepatitis and malaria are the focus of this review, we will discuss how host genetic factors are associated with severity, drug response and/or drug-induced liver injury in the context of viral hepatitis and malaria.

### 2.1. Hepatitis B Virus (HBV) Infection

Chronic HBV infection is a leading cause of morbidity and mortality worldwide. More than 250 million people were affected by chronic HBV infection in 2015 [28]. Given the association of chronic HBV infection with the progression to hepatocellular carcinoma (HCC), the incidence of HCC related to HBV reportedly increased between 1990–2015 [30]. In clinical practice, most antiviral drugs are nucleotide analogs (tenofovir) or nucleoside analogs (lamivudine and entecavir) that inhibit HBV genome amplification [31]. Despite the availability of effective antiviral drugs, current therapies do not inhibit *de*
*novo* formation of extrachromosomal DNA (covalent closed circular DNA or cccDNA), resulting in persistent infection in host hepatocytes and subsequent reactivation. Moreover, none of the antiviral therapies achieve a radical cure, as indicated by a complete loss of hepatitis B surface antigen (HBsAg) in blood [32]. Thus, an improvement in antiviral drug efficacy is inevitable and a human-relevant disease model that facilitates the assessment of HBV persistence and a radical cure is needed. The detection of HBsAg in serum is a standard diagnostic assay for acute HBV infection (within 1–10 weeks after infection) [33]. Thus, HBV infection modeling using the liver-on-a-chip should allow assessments of HBsAg and cccDNA, a molecular marker of the chronicity and persistence of viral hepatitis In Vivo.

Host genetic factors play a crucial role in the severity and drug response of HBV infection. Many single nucleotide polymorphisms (SNPs) are associated with diverse clinical outcomes of hepatitis B infection among individuals, varying from asymptomaticity, chronicity, and immunologic unresponsiveness to vaccine or the risk of HCC. These genetic variants are mainly identified in human leukocyte antigen loci, which are relevant to host immune responses. For more details of those SNP loci, the review by Akcay et al. [34] is highly recommended. Regarding the personalized treatment of HBV infection, the appropriate period to initiate therapy depends on the risk of disease progression, cirrhosis and HCC, viral replication status, and liver disease stage. Criteria for selection of the drugs rely on the likelihood of the host response to the drug, which is conventionally determined based on ALT levels, HBV DNA or virus genotypes. Finally, the decision to stop treatment requires monitoring of hepatitis B envelope antigen in blood [35]. Based on these clinical guidelines and previous reports, the liver-on-a-chip possibly provides a platform to predict drug responses by measuring HBV DNA and persistent infection by assessing cccDNA [36]. Moreover, single nucleotide polymorphisms associated with tumorigenesis may be useful for determining which patients who are at risk of HCC should receive preventive therapy [37].

### 2.2. Hepatitis C Virus (HCV) Infection

Hepatitis C virus causes acute and chronic hepatitis, an inflammation of the liver. The World Health Organization estimated that 71 million people worldwide live with chronic hepatitis C virus infection. However, the chronicity of HCV infection significantly leads to the development of liver fibrosis, cirrhosis or hepatocellular carcinoma. The majority of liver cancer cases is caused by hepatitis C. In 2016, approximately 400,000 people died from cirrhosis and hepatocellular carcinoma. Given the more than 95% efficacy of antiviral therapy, the risk of death caused by cirrhosis and liver cancer can be reduced (www.who.int/news-room/fact-sheets/detail/hepatitis-c, accessed on 11 June 2021).

Treatments for HCV infection rely on interferon (IFN)-based regimens. Given the life-threatening side effects of IFN, HCV treatments are ineffective in at least half of all patients [38]. The addition of the antiviral drug ribavirin (nucleoside analog) significantly enhances cure rates; however, interferon–ribavirin-based treatments are not well tolerated, emphasizing the need for a safer and effective drug. Given the key role of viral proteases in processing a polyprotein precursor encoded in the HCV RNA genome, viral proteases have become common drug targets (direct-acting antiviral agents or DAAs) to block host entry and RNA replication. As a supplement to the combination of ribavirin and pegylated interferon alpha, DAAs increase therapeutic efficacy to approximately 100% in genotype 1-infected patients [39,40], the most prevalent strain globally. Combinations of DAA are more effective, in the absence of the administration of ribavirin or interferon. Despite these effective therapies, many concerns related to drug resistance [41], the high cost [39], and global accessibility of these new antiviral agents remain.

Responses to HCV therapy vary among populations. Following treatment with the combination of IFN plus ribavirin, the sustained virological response of East Asians was reportedly greater than that of patients of European ancestry [42,43]. African and Latino people exhibit a poorer response to HCV therapy than Caucasians [44,45,46]. An association between polymorphisms in the *IL28B* gene and sustained virological response, the standard indicator of successful treatment, has been documented among African Americans and those of European ancestry [5].

Although pegylated interferon-alpha combined with ribavirin is the most effective treatment, many patients with chronic HCV infection do not respond to it [38]. Gene transcription profiles of liver biopsies of the responders reveal a predominance of IFN-inducible genes [47]. Hwang et al., identified DNA polymorphisms associated with responsiveness to combination therapy among Taiwanese patients with chronic HCV infection. Regarding the identified SNPs, the outcome of HCV combination therapy is potentially predictable [48]. In a prospective cohort study, a SNP variant in the promoter region of IFN-alpha was associated with the response to IFN therapy in African and Caucasian American patients with chronic HCV infection [49]. In addition to the IFN-alpha gene, genetic polymorphisms in the IFN pathway are reportedly associated with the response to therapy for European–American patients with chronic HCV infection [50]. Thus, the right drug for individuals must be selected [51].

For DAAs, limited data are available on pharmacogenomic-based treatment outcomes compared to the abovementioned combination therapy. In cohort studies of specific populations, DAAs were effective (based on sustained viral rate) and safe for patients with chronic HCV infection [52,53,54]. Notably, these outcome data were obtained for the real-world use of DAAs; no control group was available for comparison. Instead of treatment outcomes, a few reports show an association of host genetic variants with plasma levels of sofosbuvir [55] and telaprevir [56], an indicator of pharmacokinetics. Sofosbuvir, an NS5B polymerase inhibitor, is a prodrug that is predominantly metabolized in the liver to produce the GS-331007 metabolite. The plasma levels of the GS-331007 metabolite are associated with SNPs in *ABCB1* (ATP binding cassette subfamily B member 1) and *HNFα* (hepatocyte nuclear factor 4 alpha) variants, encoding sofosbuvir transporters. Drug interactions among the DAAs were also observed [57]. HCV infection is frequently observed in individuals with HIV infection or recipients of solid organ transplants. Thus, the drug–drug interaction of DAAs and anti-HIV or immunosuppressant drugs may need further elucidation. Taken together, these findings clearly suggest that host genomics plays a major role in the antiviral response and provide a new approach for the individualization of therapy in patients with HCV chronic infection.

### 2.3. Malaria

Malaria is a major global infectious disease that poses a risk to hundreds of millions of people, especially in Africa. The causative organisms are intracellular protozoan parasites, including *Plasmodium* *falciparum*, *P. vivax*, *P. ovale*, *P. malariae*, and zoonotic *P. knowlesi*. Among *Plasmodium* species, *P. falciparum* is the most virulent, while *P. vivax* has the widest geographical distribution. In 2019, an estimated 229 million cases of malaria were reported, with more than 90% of cases in Africa and 409,000 deaths worldwide. In addition to Africa, Southeast Asia and the eastern Mediterranean zone are malaria endemic areas (World Malaria Report, WHO 2020).

Malaria parasites require multiple niches in the human host, including the liver and erythrocytes, to grow and survive. Liver-stage malaria is asymptomatic; however, it is a bottleneck of the symptomatic phase of intraerythrocytic malaria. Briefly, when a female *Anopheles* species bites a host, sporozoites circulate in the bloodstream. In the liver, sporozoites invade hepatocytes via Kupffer cells or sinusoid endothelial cells. In hepatocytes, sporozoites undergo several rounds of cell proliferation, generating thousands of merozoites. Intrahepatic development of *Plasmodium* parasites causes no clinical manifestations. After a week, rupture of infected hepatocytes, known as liver schizonts, releases merozoites into blood circulation, where they invade erythrocytes to start the intraerythrocytic life cycles: ring-form trophozoites, trophozoites, and schizonts. Merozoites are released from erythrocytic schizonts and infect other erythrocytes. The release of merozoites from erythrocytic schizonts causes periodic fever. In contrast to intraerythrocytic development, some *P. vivax* or *P. ovale* sporozoites undergo a non-proliferative, metabolically inactive stage known as hypnozoites, which may be reactivated months or years after sporozoite infection [58]. Thus, hypnozoites are the cause of clinical relapse, and the majority of *P. vivax* cases are hypothetically from liver-dwelling hypnozoites [59]. Thus, the *Plasmodium* liver stage represents a target for blocking progression to clinical malaria and for preventing relapse in subjects with *P. vivax* and *P. ovale* malaria. Current prophylactic regimens are atovaquone and proguanil, targeting liver-stage schizonts. Only primaquine and tafenoquine possess anti-hypnozoite activity [60].

Despite radical cure and relapse prevention, host genetic factors influence clinical outcomes. In phase 3 clinical trials, primaquine and tafenoquine caused hemolysis in G6PD-deficient patients [61,62]. Moreover, the isoenzyme cytochrome P450 2D6 (CYP2D6) present in hepatocytes is responsible for metabolizing primaquine, generating 5-hydroxyprimaquine as a metabolite [63]. Given the instability of 5-hydroxyprimaquine, nonbiologically active 5,6-ortho-quinone is deployed as a surrogate marker of 5-hydroxyprimaquine [64]. Genetic polymorphisms in the hepatic CYP2D6 possibly impair the metabolism of primaquine, leading to the failure of a radical cure and potential relapse [65]. Different CYP2D6 genotype alleles are classified according to enzyme activity, ranging from poor to ultrarapid metabolizers [66]. Several studies conducted in malaria endemic areas have observed an association between impaired CYP2D6 activity and relapse [67,68,69]. In contrast to the abovementioned studies, the allele-classified CYP2D6 activity score did not contribute to relapse of *P. vivax* malaria in a cohort of Australian Defense Force personnel deployed to Papua New Guinea and East Timor [70]. Thus, the association of CYP2D6 and relapse in malaria endemic areas is difficult to prove. Among populations who are at risk of *P. vivax* malaria, a G6PD deficiency or CYP2D6 impairment causes primaquine ineligibility, representing approximately 35% of patients [71]. Collectively, these unsolved, well known situations have highlighted the need for safer and more effective therapeutics against *Plasmodium* parasites in the liver [71].

## 3. Drug-Induced Liver Injury (DILI) in Personalized Medicine

Drugs and drug metabolites are capable of damaging the liver, resulting in a spectrum of symptoms ranging from mild to acute liver failure. The most common drugs inducing liver injury are antimicrobials [72,73,74]. Despite the rare incidence and uncommon cause of acute liver failure, DILI poses a threat to morbidity and mortality. This complication exerts a substantial effect on the post-marketing of newly approved drugs, especially DILI-causing drugs [75].

Evidence of drug-induced liver injury in clinical trials of new drugs for chronic liver disease (cirrhosis from hepatitis C virus, hepatitis B virus, or nonalcoholic steatohepatitis) warrants screening for potential adverse side effects, especially for patients with advanced chronic liver diseases with a higher risk of mortality [76,77,78]. Current guidelines of the FDA recommend measuring the levels of alanine aminotransferase, aspartate aminotransferase, total bilirubin, and alkaline phosphatase as surrogate markers of potential DILI as criteria to stop an ongoing clinical trial of an investigational drug [75].

The mechanism underlying direct and indirect DILI is well understood; however, the pathogenesis of idiosyncratic injury remains unknown [75]. Genome-wide association studies have identified specific alleles associated with idiosyncratic injury. These genetic marks are found in the HLA class I and II alleles of the major histocompatibility complex or outside the MHC region (an immunomodulatory gene encoding PTPN22). Nevertheless, genotyping of DILI-associated HLA alleles has not yet been deployed in clinical studies. Three alleles associated with DILI caused by antimicrobial agents have been identified: flucloxacillin (skin infection) [79], amoxicillin/clavulanic acid [80] and terbinafine (pityriasis versicolor and fungal nail infections) [81].

## 4. Basis of the Liver Microarchitecture for the Liver-on-a-Chip Design

The microarchitecture of the smallest unit of the liver is important for the design to recapitulate the liver-on-a-chip device (Figure 1). As the essential units of multicellular organisms, this section discusses various cell types and their surrounding microenvironments crucial for liver functions. The two lobes of a human liver are grossly subdivided into eight segments. Each segment is composed of lobules and hexagon-shaped blocks (Figure 1A, middle subpanel). The center of the liver lobule is the central vein, and the apex of the hexagon consists of a group of three vessels: the hepatic artery, portal vein and bile duct, collectively called the portal triad (Figure 1A, right subpanel). Distinct types of cell surround the central vein in a radial pattern [82] (Figure 1B). In the hepatic portal vein, a single layer of sinusoidal endothelial cells (SECs) lines the vein, and each has small cytoplasmic holes, called fenestrations, to allow noncellular fluid to reach hepatocytes on the basal side (Figure 1C). The area between SECs and hepatocytes is the perisinusoidal space, termed the space of Disse, where microvilli of hepatocytes are exposed to noncellular components of blood (Figure 1C). Gastrointestinal tract-derived, nutrient-enriched blood flows into the liver through the hepatic portal vein and drains into the central vein. Thus, nutrients and oxygen are diffused across fenestrated SECs to the basolateral side of hepatocytes in a gradient [83]. The apical side of adjacent hepatocytes forms bile canaliculi, where bile acid is secreted and transported to the bile duct at the portal triad.

Cell polarization is important for the directional transport of molecules across the basolateral and canalicular membranes. In liver lobules, liver cells are grouped into parenchymal and nonparenchymal cells. Hepatocytes are the major parenchymal epithelial cells, constituting 60% of the total number of liver cells [84]. Absorption of LDL occurs at the basal domain via the LDL receptor, while secretion of bile salt occurs at the apical domain via ATP-binding cassette proteins. Tight junction proteins, such as ZO1, prevent the paracellular flux of molecules. Extracellular matrixes act as scaffolds for hepatocyte adhesion. Unlike the basal lamina (layer of extra cellular matrixes), hepatocytes are surrounded by laminin, collagen type IV and fibronectin [85]. Another cell type is nonparenchymal cells that function in maintaining the liver microstructure and other functions, including sinusoidal ECs, hepatic stellate cells (vitamin A storage and collagen production), Kupffer cells (resident macrophages) and cholangiocytes (bile acid secretion).

## 5. Liver-on-a-Chip for Hepatotropic Infectious Diseases

Sodunke et al. first generated a single microchannel for the culture of primary rat hepatocytes or HepG2 hepatocellular carcinoma cell lines [86], without fluid flow, in order to model HBV infection. Hepatic cells grew on collagen-coated microchannels, where the culture medium was filled via capillary force to minimize shear force. The chip was composed of PDMS and contained various forms of microchannels, including rectangles, hexagons, and circles. Unlike HepG2 cells, culture of primary hepatocytes is complicated. Briefly, primary hepatocytes were prepared by perfusing the liver of 7- to 10-week-old rats with a buffer containing collagenase. Cells were seeded into the microchannel using a pipette to minimize shear stress caused by suction. Rat hepatocytes were purified using gradient centrifugation and cultured with medium supplemented with an insulin-transferrin-selenium mixture, epidermal growth factor and hydrocortisone. Due to the optical transparency of the PDMS device, cell morphology was observed under a microscope. Since primary hepatocytes undergo dedifferentiation in culture In Vitro, the hepatocyte differentiation status must be validated. Cultured cells were scraped from the microchamber and subjected to a gene expression analysis. At day 4 after cell culture, the rat hepatocytes expressed hepatocyte-specific hepatic nuclear factor-4 and albumin transcripts. Moreover, an HBV genome-carrying adenovirus infected the majority of cultured rat hepatocytes, resulting in the release of HBV DNA in the culture medium. The same group generated a liver sinusoid-on-a-chip to mimic blood flow and the fenestration of sinusoidal endothelial cells lining the liver lobules. Kang et al., deployed a dual microchannel separated by a microporous membrane layer with a pore size of 0.4 µm. The microchannel was composed of PDMS, while the microporous membrane was composed of polyethylene terephthalate or PET. Because human liver tissues and cells are available for scientific investigators, the study obtained primary hepatocytes from the Liver Tissue Cell Distribution System (LTCDS), a National Institutes of Health (NIH) service contract. Primary human hepatocytes (PHHs) and immortalized bovine aortic endothelial cells were cultured on opposite sides [87] (Figure 2A). Similar to the study by Sodunke et al., HBV infected hepatocytes cultured in a liver sinusoid-on-a-chip, releasing HBV DNA in the culture medium. Altogether, these results imply the validity of the device model to mimic the In Vivo condition. HBV DNA in culture medium was detected using PCR in both studies, indicating viral replication in these 2D-based cultures. Notably, HBsAg and cccDNA levels were not examined.

Ortega-Prieto et al., developed a microfluidic platform for the 3D culture of PHHs [36]. With support from the collagen-coated polystyrene scaffold, PHHs grew in three dimensions, formed bile canaliculi, and underwent cell polarization (hepatic microvilli). The 3D PHH cultures were susceptible to HBV infection, resulting in the release of HBsAg and HBV DNA. Moreover, episomal cccDNA was detected in this culture device, allowing persistent HBV infection. Importantly, innate immune responses of the HBV-infected 3D PHH models were similar to those observed in individuals infected with HBV. Thus, this model is suitable for investigating host factor-dependent mechanisms underlying immune evasion of the virus. Nevertheless, the abovementioned liver-on-a-chip devices do not examine the development of HCC, due to the requirement for long-term culture.

## 6. Biosensor Assays of Liver Function

Biosensors have advanced disease diagnosis and bedside monitoring and are also known as lab-on-chip sensors. This section revisits current biosensors designed to measure liver functions. Briefly, the three main components of biosensors are recognition, transduction and readout. In terms of recognition, biological molecular targets bind to receptors on transmissible surfaces. Following a biochemical interaction, the target-bound receptor causes a change on the surface, which is translated into a quantifiable, dose-dependent physical signal., Many forms of receptors have been immobilized on the transducer surface, such as antibodies, enzymes, peptides, nucleic acids, aptamers or cells, collectively called biomolecular probes. Biochemical changes on the surface may be mechanical, i.e., molecular interactions, or electrochemical, i.e., electron transfer. Thus, the transducer translates those changes into readable and interpretable signals.

In a clinical laboratory, liver function testing is a common routine assay for evaluating liver dysfunction or damage caused by infections (hepatitis viruses), cirrhosis, steatosis and drug-induced hepatitis. Biomarkers of liver function include albumin, bilirubin, alanine aminotransferase (ALT), and aspartate aminotransferase (AST) in blood. Biosensors equipped with microchips are grouped into impedimetric and ampero-metric biosensors. The impedimetric biosensor immobilizes recognition molecules onto an electrode surface. An electrical impedance signal is proportional to analyte activity. The ampero-metric biosensor relies on biochemical oxidation or reduction, generating electron transfer as a signal., Table 1 summarizes the current biosensors used for assessing hepatocyte functions and viability.

Song et al. developed a three-in-one biosensor device to quantitatively measure the concentrations of cholesterol, bilirubin, ALT and AST. The device consists of immobilized enzymes on nano-porous silicon attached to the working electrode to increase the surface area for target recognition. As a byproduct of the enzymatic reaction, H_2_O_2_ is oxidized to generate two electrons, which are then transferred to OH on the electrode to transduce an electrochemical signal. Due to the stability of enzymes, the device should be stored at 4 °C when not in use. The maximum period of enzymatic stability is 120 days [88].

Human serum albumin (HSA), the most abundant protein circulating in plasma (60% of the total serum proteins), is synthesized by hepatocytes and secreted into the blood at a concentration of 35–50 mg/mL in healthy individuals [92]. Albumin detection primarily relies on the use of antibodies. Chuang et al. immobilized an anti-human serum albumin antibody onto a glass surface adjacent to two Au electrodes. Binding of albumin and anti-human albumin causes impedance between two adjacent Au electrodes that is detected by an electrochemical impedance spectroscopy system under alternating current conditions. The increase in impedance depends on the concentration of human serum albumin, with the lowest detection limit of 2 × 10^−4^ mg/mL [89]. Huang et al. developed a microfluidic device to detect albumin levels in urine [90]. The device was equipped with three electrodes: a working electrode (Au or gold), a counter electrode (Pt), and a reference (Ag or silver) electrode. As an indication of electrochemical activity, the redox reaction of $\mathrm{Fe}{\left(\mathrm{CN}\right)}_{6}^{3^{-}}$/$\mathrm{Fe}{\left(\mathrm{CN}\right)}_{6}^{4^{-}}$ generates electrons, resulting in cathodic peaks and anodic peak currents. Given the disulfide or thiol bonds among the amino acids of HSA, sulfur–gold bonds are potentially formed, resulting in strong covalent binding of albumin to the Au electrode. Since HSA binding interferes with the redox reaction mentioned above, the change in the ratio of the cathodic and anodic peaks between the reference Ag electrode and the HSA-bound Au electrode are detected and are proportional to the HSA concentration. Notably, pH affects the binding of albumin to the Au electrode; thus, pH adjustment is required before use.

In addition to antibody- and electrode-based biosensors, a fluorescence probe was recently developed [93]. The probe was capable of binding to major proteins in biological fluid, such as globulin, fibrinogen and transferrin. However, it tended to specifically and selectively bind HSA and was applicable for detecting HAS in artificial urine samples. Nevertheless, validation of the probe with biological samples and a method to fabricate it as a biosensor are still needed.

## 7. Combination of a Biosensor and Organ-on-a-Chip

The key features of biosensor-integrated organs-on-chip are the real-time monitoring of cell functions and viability. Kinetic data allow us to dissect the mode of action of cytotoxic substances. In this section, current organs-on-chips integrated with biosensing microdevices are highlighted, especially those applicable for assessing liver integrity and injury.

### 7.1. Cellular Barrier Function

Selective transportation of substances and biomolecules via apical-basolateral compartments of many cells and tissues relies on the integrity of the epithelial or endothelial cell layer. Maoz et al. combined a dual biosensing system consisting of multielectrode arrays (MEAs) and electrodes for transepithelial electrical resistance (TEER) to measure the electrical activity of cardiomyocytes and ionic conductance between the paracellular space, respectively [94] (Figure 2B). Cardiomyocytes were derived from human iPSCs, while vascular endothelial cells were isolated from the human umbilical cord. In the presence of the proinflammatory cytokine TNF-α, the endothelial barrier was disrupted, including the formation of a gap between adjacent cells and cytoskeletal changes, resulting in a decrease in TEER. In addition, the MEA was able to detect an increase in the beating rate when cardiomyocytes were directly exposed to isoproterenol. Thus, electrode-based biosensors are useful for the assessment of endothelial barrier function. Alternatively, an electromechanical biosensor composed of biocompatible carbon black nanotubes and thermoplastic polyurethane was embedded as a cantilever layer. Thus, this device is noninvasive for cardiomyocytes and provides an electronic readout of cell contraction under physiological conditions [95].

### 7.2. Cell–Cell Communication via Paracrine Signaling

Alcohol causes liver fibrosis, which is characterized by an excess amount of collagen and other matrix proteins produced in a repair process after injury or inflammation; however, the underlying mechanism remains unclear. TGF-β is a pleiotropic cytokine that reportedly induces fibrous tissue formation during the recovery phase of liver injury [96]. Zhou et al. developed a microfluidic device composed of two microchambers for the coculture of hepatocytes and hepatic stellate cells and an additional three microchambers for detecting TGF-β levels. Similar to the liver lobule, the coculture chambers allow both cell types to be in close proximity. The three TGF-β detection microchambers contained TGF-β-specific aptamer electrodes. Binding of TGF-β to the aptamer changed the conformation and thus reduced the electrochemical signal of the redox reporter attached to the aptamer. The TGF-β-detecting microchamber is located side-by-side with the cell culture microchamber. Thus, the device was able to detect the source of TGF-β and the direction of cell–cell communication. Briefly, alcohol first induced hepatocytes to release TGF-β, which subsequently diffused and activated hepatic stellate cells to secrete TGF-β [97]. Identification of the source of TGF-β and responding cells potentially provides an opportunity to prevent or ameliorate liver fibrosis [98].

### 7.3. Cell Viability

Cell survival in a microchamber of a microfluidic chip is critical for assessing biological responses to stimuli. Oxygen represents mitochondrial respiration in a metabolically active cell. Thus, its amount indicates viable cells. The use of oxygen microsensors can lead to the discovery of the mode of action of drugs against hepatocytes. Prill et al. developed a microchip device for long-term maintenance of spheroids of HepG2/C3A cells, while phosphorescent microprobes were embedded in the spheroid to detect cell viability and assess the hepatotoxicity of drugs in the liver-on-a-chip after repeated dosing [98]. Real-time tracking of oxygen levels enabled the assessment of the onset of cytotoxicity, as well as the reversible or irreversible effects of amiodarone (antiarrhythmic drug) and acetaminophen (pain and fever reliever) in a dose-dependent manner. Importantly, this device allows microscope-based optical measurements of oxygen levels due to its transparency and the lack of focus interference. Moreover, the phosphorescence dye loaded in polystyrene microbeads (10–50 µm in diameter) provided kinetic information on oxygen levels in 3D-cultured hepatocellular carcinoma cells and primary mouse hepatocytes upon exposure to acetaminophen. Two possible modes of action of acetaminophen were proposed: one dependent on and another independent of a toxic acetaminophen metabolite [99].

## 8. Source of Hepatocytes for the Personalized Liver-on-a-Chip

A personalized liver-on-a-chip requires patient-specific cells. In this section, sources of patient-specific cells and their limitations in individualizing the liver-on-a-chip are reviewed. Since the microchamber requires a very small number of cells, the scalability of cells is less problematic. However, in the case of medium- or high-throughput screening, expansion of the original source of cells prior to hepatocyte induction is necessary (Figure 3).

### 8.1. Liver Biopsy

Biopsied tissue could be obtained during a pathological examination or biobank analysis. Given the small size of the cell culture chamber in a chip, a sufficient number of cells can be obtained. However, the number of donors is limited, preventing accessibility to some rare genetic variants. Additionally, performing liver biopsy in healthy individuals carries greater risks than benefits.

### 8.2. Trans.-Differentiation

Since biopsy of the human liver is too invasive, an alternative method is to generate patient hepatocytes from other cell types obtained through noninvasive or less invasive procedures. Generally, cell differentiation is a hierarchical process in which pluripotent stem cells or multipotent stem cells differentiate into somatic cells with specific functions (mature somatic cells). In contrast, cell trans-differentiation is the process in which mature somatic cells undergo transformation into distinct types of mature somatic cells without passing through pluripotency (i.e., induced pluripotent stem cells) or multipotency (i.e., adult stem and progenitor cells). Thus, the process to generate personalized hepatocytes for In Vitro assays is shorter.

#### 8.2.1. Mesenchymal Stem Cells (MSCs)

MSCs are a type of adult stem cell with the ability to differentiate into several cell types, including hepatocytes. Several methods have been developed to induce MSCs to undergo trans-differentiation [100]. In addition to their differentiation ability, MSCs are highly proliferative In Vitro in the presence of growth factors [101]. Thus, their expansion before trans-differentiation is feasible, if needed. Adipose tissue is a source of MSCs. Adipose tissue-derived MSCs are capable of differentiating into hepatocytes [102,103]. Unwanted fatty tissue obtained after liposuction or plastic surgeries (Jurgens et al., 2008) and dental pulp (Klingemann et al., 2008) are additional sources of MSCs.

#### 8.2.2. Fibroblasts

Fibroblasts are nonparenchymal cells located in connective tissue, including skin. Unlike MSCs, trans-differentiation of fibroblasts into hepatocytes requires the ectopic expression of a set of three transcription factors: endoderm-specific FOXA2, hepatocyte-specific HNF4α, and C/EBPβ together with the cell proliferation-enhancing factor cMyc [104]. Fibroblast-derived hepatocytes are able to synthesize and secrete albumin, possess cytochrome activity and store glycogen, which are hallmarks of hepatocyte function. Fibroblasts obtained from skin biopsy grow on collagen type I-coated plates supplemented with growth factors [105]. Notably, fibroblast-derived hepatocytes function similarly to hepatocytes, but they also undergo dedifferentiation, a phenomenon in which hepatocytes lose their functions.

#### 8.2.3. Hematopoietic Cells

##### Hematopoietic Stem and Progenitor Cells

Evidence suggests that hematopoietic cells are capable of differentiating into hepatocytes. Khurana et al. documented that the sera of mice with chemical-induced liver injury induce the differentiation of mouse bone marrow (BM)-derived hematopoietic stem cells into hepatocyte-like cells, implying the requirement for soluble host factors relevant to liver recovery [106]. Later, the same group reported that overexpression of hepatocyte nuclear factor (HNF)-4α in BM-derived hematopoietic cells (negative for lineage markers and positive for onco-statin-M receptor b) generates hepatocyte-like cells capable of synthesizing albumin and active cytochrome p450 enzyme [107]. Sellamuthu et al. reported that umbilical cord blood-derived hematopoietic stem cells (HSCs) differentiate into hepatocyte-like cells [108]. In the absence of ectopic expression of transcription factors, the cells were cultured with fibroblast growth factor 4 (FGF4) and hepatocyte growth factor (HGF). By day 14, hepatocyte-like cells were microscopically observed, and albumin and α-fetoprotein were detected in the culture medium. Moreover, human BM-derived mononuclear cells differentiated into hepatocyte-like cells expressing cytoplasmic albumin [109]. However, researchers have not determined whether human hematopoietic cell-derived hepatocytes are capable of metabolizing drugs. The use of BM and CB as sources of personalized liver-on-a-chip remains a challenge due to the invasiveness of the procedure to obtain HCs.

##### Monocytes

Monocytes are mononuclear phagocytic cells circulating in blood. Human monocytes reportedly differentiate into hepatocyte-like cells capable of metabolizing phase I and phase II drugs. Thus, monocyte-derived hepatocyte-like cells were deployed to assess drug-induced liver injury. Due to the less invasive nature of their collection from peripheral blood, monocytes are suitable for personalizing the liver-on-a-chip of individuals [110,111].

### 8.3. Human Induced Pluripotent Stem Cells (iPSCs)

Due to the invasiveness of cell harvesting procedures and a limited proliferation capability outside the body of primary cells, pluripotent stem cells are an alternative source to overcome these limitations. Following the ectopic expression of four transcription factors, somatic cells are reprogrammed into pluripotent stem cells capable of proliferating and differentiating into various cell types [112]. Thus, patient-specific iPSCs are widely generated and deployed for personalized disease models [113,114], drug screening assays [115] and cell and tissue regeneration [116,117]. Various types of somatic cells have been induced to enter a pluripotent state, including skin fibroblasts, mesenchymal cells, and hematopoietic cells [118]. Our group reported that human iPSCs derived from hematopoietic CD34+ cells differentiate into hepatocytes in a 2D culture and generate hepatic organoids [119]. Given the high proliferation ability of iPSCs, a large number of cells of interest can be obtained, allowing high-throughput utilization. However, some types of iPSC-derived target cells functionally mimic cells of the fetus but not adults. Thus, the phenotypes to be examined must be confirmed prior to use, i.e., cytochrome P450 2D6 activity in hepatocytes for primaquine metabolism. Moreover, the source of somatic cells is critical for efficient cell differentiation due to epigenetic memory of cell origins [120]. Thus, the state of pluripotency must be assessed, particularly epigenetics such as DNA methylation. A number of human iPSC-derived cells have been utilized in the form of organs-on-chips for the myocardium [121,122], spinal cord, skeletal muscle [123], and cardiomyocytes [124].

## 9. Opportunities and Challenges in the Use of the Liver-on-a-Chip in Personalized Medicine

Various designs of the liver-on-a-chip have allowed researchers to model noninfectious pathological conditions, such as the accumulation of lipids and DILI, as well as to assess drug metabolism. This section highlights key features of those designs, which are potentially applicable for infectious diseases.

### 9.1. Artificial Porous Layer

Lee et al. developed a microchamber bordered by a parallel microfluidic channel (Figure 4A). Primary human hepatocytes grew in the microchamber without a coat of matrix protein. The culture medium flowed into the microfluidic channel and across the microfluidic endothelial-like barrier through pores 1–2 μm in diameter. The device was applied to assess the metabolism-mediated toxicity of diclofenac, a nonsteroidal anti-inflammatory drug. Given the transparency of polydimethylsiloxane (PDMS), the viability of drug-exposed hepatocytes was monitored using a fluorescence probe and microscope [17]. Later, Gori et al. slightly modified the device by increasing the size of the microchamber for culturing a greater number of cells. HepG2/C3A cells were cultured with a combination of long-chain fatty acids to induce steatosis, a condition of excess lipids in hepatocytes. Intracellular lipids and oxidative stress were then examined using a fluorescence probe, allowing the assessment of nonalcoholic fatty liver disease In Vitro [125].

Banaeiyan et al. developed a bilayer device consisting of a hexagon-like chamber located in the lower part and a tunnel network for seeding cells and feeding culture medium located at the top. Hepatocytes were perfused in the top layer and grew in the lower chamber. Nutrients and drugs entered the system through the top layer and spread radially through hepatocytes in the lower chamber. Small channels (2 μm wide and 2 μm high) were placed between the flowing medium and hepatocytes in the lower chamber to mimic the fenestration of endothelial cells. Thus, nutrients and oxygen diffused to hepatocytes in a gradient manner, and the device reduced shear stress to hepatocytes by approximately 80 times inside the chamber (5 × 10^−4^ dyne cm^−2^), which was lower than that in the diffusion channels (0.01 dyne cm^−2^). Similar to the central vein of the liver lobule, the outlet of the upper layer allows collecting culture medium to pass through hepatocytes. Due to the high optical transparency of polydimethylsiloxane (*PDMS*), the device enables an assessment of cell viability, particularly serving as a microscope-based biosensor of oxygen. Each compartment of the hexagonal chamber provides space for hepatocytes to grow in three dimensions, enhancing cell–cell interactions in all directions [126].

Considering the utility of this device for personalized medicine, human iPSC-derived hepatocytes were able to grow and secrete albumin and urea. In addition, a network of bile canaliculi, a 1–2-μm-wide bile-collecting space between the apical membranes of adjacent hepatocytes, was observed, allowing an assessment of DILI based on bile secretion. A challenge is to seed multiple cell types in the cell chamber. According to Takebe et al. human iPSC-derived hepatic endoderm, mesenchymal cells and endothelial progenitors self-organize into α-fetoprotein-expressing hepato-blasts surrounded by endothelial cells. However, further studies are needed to elucidate whether a combination of various cell types is capable of generating functioning units of the liver as a lobule [126].

### 9.2. Endothelial Cells

Attempts to fabricate a microstructure of the lobule rely on patterning hepatocytes and ECs in radial lines protruding from the central vein. Ho et al. applied an electrode array for sequential trapping of HepG2 cells and human umbilical vein endothelial cells [127] (Figure 4C). Similar to the liver lobule, the device promotes flow of culture medium in the direction of the portal vein-to-central vein axis. Based on the fluorescence assay, more than 90% of HepG2 cells and HUVECs survived. Compared to nonpatterned HepG2 cultures, the fluorescence-based enzymatic activity of CYP450-1A1 was increased.

### 9.3. Multiple Cell Types

Attempts to fabricate microfluidic devices and microarchitectures of tissue are hallmarks of chip technology. Evidence suggests that a liver-on-a-chip is capable of recapitulating the In Vivo microenvironment architecture and that the functions resemble those of human tissues compared to conventional two-dimensional (2D) cell culture. Multiple factors appear to contribute to the physiological relevance of the microfluidic chip, such as the microfluidic system and cellular interactions with other cells or matrix proteins. Given the continuous flow of culture medium, a liver sinusoid-on-a-chip enhances human hepatocyte secretion of albumin and urea to levels higher than those of a static device [128]. As a 3D scaffold, heparin-coated micro-trenches and perfused culture reportedly support the growth of primary mouse hepatocytes (PHHs) capable of secreting albumin and urea for up to four weeks [129]. Moreover, the liver organ chip has been used to evaluate the biochemical activity of enzymes involved in phase I and II metabolism in the liver, facilitating an assessment of drug metabolism [128].

### 9.4. Integration of Biosensors

In this section, we briefly provide a basis and discuss in-depth the use of TEER and MEA as assessments of barrier integrity and cellular characteristics. Some examples of electric biosensor-integrated organs-on-a-chip are provided.

#### 9.4.1. Cellular Barrier

Transportation of substances and biomolecules plays a role in maintaining organ functions, including the brain [130] and liver [131]. Selective transportation of these molecules via the apical-basolateral axis of many organs mainly relies on intercellular junctions of epithelial and endothelial cells. The TEER electrode directly measures electrical resistance between two cellular compartments and is not interfered with by macromolecule transportation. Thus, the TEER electrode enables real-time measurements of cellular barrier integrity in a physiological context. A number of studies have reported the integration of TEER electrodes with the organ-on-a-chip. This section briefly summarizes the principle of TEER and highlights the use of TEER as a biosensor of cellular barrier integrity in a microfluidic chip. For those interested in electronic engineering of TEER electrodes, we recommend a review by Srinivasan et al. [132].

##### Basis of TEER

Briefly, a classical method to measure electrical resistance between two compartments is initially based on the use of two electrodes for sensing voltage and current. A well is placed inside in a larger chamber to sense the resistance of the two compartments. This inner well, called the trans-well here, has a permeable filter or membrane at the bottom to which a layer of endothelial cells attaches. Thus, the culture medium inside the trans-well is separated from the medium outside the inner well. Electrodes are placed inside and outside the trans-well, a configuration similar to apical and basolateral sides, respectively. Upon applying a direct current voltage to the electrode, the electrical resistance is then calculated based on Ohm’s law. The lack of resistance of the cell-free inner well and presence of resistance across the upper and lower chambers of the cells growing in the inner well yield a cell-specific resistance. Given damage to the cell and electrode, an alternating current can be used instead of a direct current.

##### Applications of TEER as a Biosensor

At present, TEER measurement devices are commercially available. An Epithelial Volt-ohmmeter (EVOM) applies an alternating current and is capable of measuring electrical resistance ranging from 1–9999 Ω. The EVOM deploys a pair of electrodes (4 mm wide and 1 mm thick), also known as a chopstick electrode. Off electrodes, a silver/silver chloride pellet and a silver electrode are used to measure the voltage and the passing current, respectively. In Vitro BBB models deployed EVOM to examine the permeability of the BBB among different human brain capillary endothelial cell lines [133]. On the other hand, an EndOhm chamber allows cell culture on the membrane in a cup inserted in the chamber. Top and bottom sides of the cell-attached membrane are connected with an electrode. Both the top and bottom electrodes are circular and composed of a voltage-sensing silver/silver chloride pellet in the center surrounded by a ring-shaped current electrode. Compared to the chopstick electrodes, a uniform current density is obtained from the EndOhm chamber [134,135]. In addition to the BBB model, the gastrointestinal tract and renal tubular barrier also utilize chopstick electrodes. A list of commercial biosensors and their use in the form of trans-wells or single wells is shown in Table 2.

In addition to the electrical resistance, electrical impedance, which uses the formula of ohms = voltage (V)/current (I), is deployed as an indicator of TEER. The electrical impedance is in opposition to a circuit providing direct or alternating electric current. If the endothelial layer is equivalent to an electric circuit, flow of electric current through cell junction proteins contributes to TEER. The Electric Cell-Substrate Impedance Sensing (ECIS) from Applied BioPhysics consists of an insulating film containing a gold electrode and a gold counter electrode located on the opposite side. Cells grow on the insulating film and eventually cover the electrode surface. When an alternative current (I) is applied at the bottom of the ECIS arrays, the cell-covered electrode surface generates potential (voltage) across the two electrodes. Thus, based on Ohm’s law (ohms = V/I), it increases impedance. In the case of a cell junction interruption, the voltage changes, leading to an alteration in the impedance. Based on the use of an ECIS-based device, Bernas et al. showed that human brain microvascular endothelial cells form a blood–brain barrier (BBB)-like structure with a high TEER. Exposure to lysophosphatidic acid decreases TEER, indicating the disruption of barrier integrity in real-time [136].

##### The Use of the TEER-Based Organ-on-a-Chip

Here, we provide examples of BBB-modeling microfluidic devices integrated with TEER as a biosensor of barrier integrity. Devices connecting the TEER electrode to the inlet and outlet of a microfluidic chip are not within the scope of this review, but these applications are comprehensively reviewed by Srinivasan et al. [132]. The BBB strictly restricts the transportation of substances from blood to the brain while selectively allowing the transportation of nutrients and the elimination of metabolites from CNS-surrounding cerebrospinal fluid [138]. Unlike liver sinusoid ECs, BBB endothelial cells completely lack fenestrated cell membranes and exhibit less pinocytosis. Lipophilic substances with molecular weights less than 500 Da enter the brain; however, most therapeutic molecules have molecular weights of 500–1000 Da [139]. Thus, a BBB-like In Vitro model is needed to evaluate the efficacy of drugs targeting neuropathological diseases.

Due to the relatively large size of the chopstick electrode, it is difficult to integrate into a microfluidic chip. Thus, immobilization of microscale electrodes within the microfluidic chip allows proximal contact with cells and reduces electrical resistance caused by the cell culture medium. Booth & Kim cultured a murine endothelial cell line with an astrocytic cell line on porous polycarbonate membranes located between two layers of PDMS to fabricate the BBB on a microfluidic chip. Voltage and current electrodes were inserted between the glass and PDMS layers. The transparency of glass allows morphological observation and cell viability measurements. Both electrodes were connected to an EVNOM epithelial volt-ohmeter to assess the integrity of the BBB. Histamine exposure decreased TEER, followed by recovery to the initial level. Thus, this system is suitable for measuring the transient response of the BBB to substances in real time [140]. Moreover, Griep et al. developed a PDMS-based, two-layer microfluidic chip separated by a trans-well membrane. An immortalized human brain endothelial cell line grew on the trans-well membrane, forming a barrier between the top and bottom chambers. Instead of oxidation-sensitive Ag/AgCl electrodes, two platinum electrodes were connected to the sides of the top and bottom chambers. Exposure to shear stress increased TEER, whereas exposure to tumor necrosis factor-alpha decreased TEER, suggesting an increase in tightness and a loss of barrier integrity, respectively [141].

#### 9.4.2. Characterization of Cells Using the MEA

Cell characteristics are surrogate markers of the cellular response to stimuli. The use of fluorescence probes or antibodies for labeling cells in a microfluidic chip has been hampered by technical difficulties and cell damage. These cell-labeling tools allow qualitative analyses at specific times; therefore, many chips are needed to examine drug toxicity in a dose- and time-dependent manner. Thus, a label-free, real-time assay would enable quantitative and continuous monitoring of cells in a chip. The MEA was used as a tool for characterizing cells and primarily relies on electric impedance, which is a measurement of electrical resistance to alternating current. Electrical impedance spectrometry has been deployed as a method for characterizing cells based on their electric properties [142,143]. Thus, various forms of electrodes have been developed for assessing the electrical impedance properties of cells, including morphology [144], growth and proliferation [145], composition of the lipid bilayer [146,147], exo/endocytosis [148], and cell movement on matrix protein-coated solid surfaces [149]. Due to the microscale cell culture in a tiny chamber, the MEA can closely measure changes at the single-cell level. Moreover, given the lack of cell labeling and noninvasiveness, MEA does not damage cells, reducing confounding factors in cytotoxicity tests. Here, we present examples of the integration of the MEA into a microfluidic device for cell assays.

Meissner et al. developed a microfluidic chip for liver toxicity assays based on morphological changes in cultured HepG2/C3A cells without cell labeling. Exposure to acetaminophen resulted in a change in the cytoskeleton, consistent with a decrease in electrical impedance. Importantly, a decrease in electrical impedance was detected at 2 h after drug exposure, a time earlier than the detection of cellular damage at 24 h after drug exposure. Moreover, the device was capable of distinguishing recovering cells from nonexposed cells using two different frequencies (10 kHz and 3 MHz). Four triangle-shaped chambers surrounded by PDMS-based micropillars, which decreased shear stress and trapped cells, were used to grow cells in the device. In each chamber, a microelectrode was designed as an interdigitated configuration to cover a large surface area. The assay detected the loss of cell–cell contact before cell death, showing greater sensitivity than the endpoint cell viability assay [150].

An increase in the electrode-covering area possibly enables the sensitive detection of electric impedance; however, attempts to precisely measure a single cell remain technically challenging. Asphahani et al. developed planar microelectrodes in which each human glial cell was patterned on gold microelectrodes via cell-peptide ligand interactions to enable real-time monitoring of single cells [151]. Cells were covalently bound with a lysine-arginine-glycine-aspartic acid peptide. Patterning individual cells on electrodes with surface areas similar to the cell size increases the detection sensitivity of electric impedance. Exposure to neuron-inducing substances decreased electrical impedance in a time- and concentration-dependent manner. Consistent with the decrease in electric impedance, morphological observations showed signs of early apoptosis, and a flow cytometry-based assay revealed apoptotic cells [152].

## 10. Conclusions and Perspectives

Given the emerging evidence of the need for precision medicine in infectious diseases, the assessment of patient-specific responses to antiviral or anti-plasmodial treatments remains a challenge. Currently, the use of the liver-on-a-chip to model hepatitis virus infection provides a useful platform for individualizing treatments; however, liver-stage malaria is still under investigation. Integration with biosensors to enhance the utility of the liver-on-a-chip allows real-time or kinetic monitoring of cell functions, leading to novel biological and mechanistic insights that might be a target for new therapeutic approaches. For instance, a combination of a dual biosensor with the liver-on-a-chip is proposed for assessing the effects of biological barriers on hepatocyte survival and functions (Figure 5). Briefly, iPS cell-derived sinusoidal endothelial cells grew on porous PET membranes, while iPS cell-derived hepatocytes grew on matrix protein-coated surfaces in the lower chamber. Based on the model proposed by Maoz et al. two different electrodes are placed at distinct positions. First, TEER is inserted in the upper and lower chambers. When gaps between endothelial cells form, they cause a decrease in the electrochemical signal, allowing an examination of sinusoid integrity. In addition, the gradient concentration of biomolecules passing through the porous membrane can be monitored. Moreover, the MEA can be inserted in the lower chamber to detect the levels of oxygen or other secreted molecules as surrogate markers of cell viability and functions.

## Figures and Tables

**Figure 1 micromachines-12-00842-f001:**
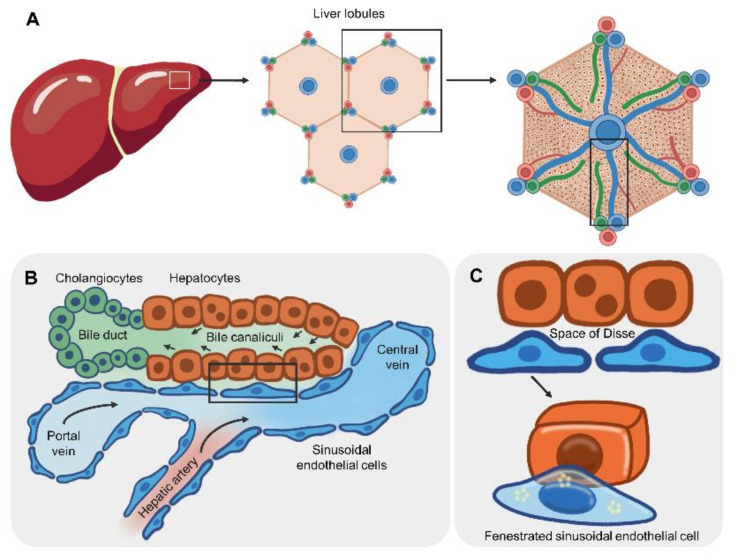
Microarchitecture and cellular compartments of liver lobules. (**A**) The liver lobule is a hexagon-shaped unit that combines with other lobules to comprise the complete liver structure. The central vein is located at the center of each liver lobule. The apical side of the hexagon consists of the hepatic artery, hepatic portal vein and bile duct (portal triad). (**B**) Hepatic cells radially surround the central vein (rectangle from panel (**A**)). Blood flows from the hepatic portal vein to the central vein and passes through a single layer of sinusoidal endothelial cells (SECs). The apical side of adjacent hepatocytes forms bile canaliculi, where bile acid is secreted and transported to the bile duct, the walls of which are lined by cholangiocytes. (**C**) Fenestration of SECs (rectangle from panel (**B**)) allows noncellular fluid to reach the perisinusoidal space (space of Disse), where the basolateral side of hepatocytes is exposed to nutrients, gases and chemical substances.

**Figure 2 micromachines-12-00842-f002:**
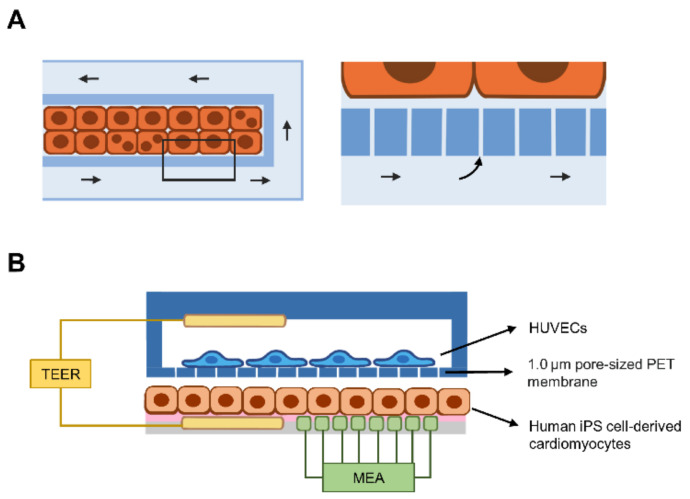
Current liver-on-a-chip models for hepatotropic infectious diseases and a combination of biosensors and organs-on-chip. (**A**) The microfluidic chip contains two compartments: a microchamber for hepatocyte culture (orange-colored rectangles) and microchannel (light blue) for fluid flow (arrows). The magnified box shows the porous layer that recapitulates the fenestration of SECs and allows the diffusion of culture medium into the cell-filled microchamber. (**B**) Coculture of human iPS cell-derived cardiomyocytes and HUVECs in double microchambers. Human primary hepatocytes (orange-colored rectangles) are grown on the fibronectin-coated surface (pink) of the lower chamber, while HUVECs are grown on the porous PET membrane of the upper chamber. The transepithelial electrical resistance (TEER, yellow) electrode is inserted into the upper and lower chambers. Gap formation between endothelial cells causes a decrease in TEER. Moreover, a multielectrode array (MEA) can be inserted in the lower chamber to detect the beating rate of cardiomyocytes. Abbreviations: HUVEC, human umbilical vascular endothelial cell; PET, polyethylene terephthalate.

**Figure 3 micromachines-12-00842-f003:**
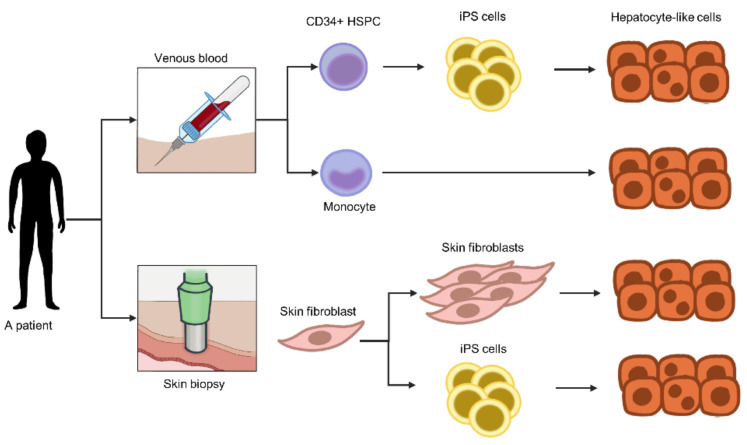
Sources of patient-specific hepatocytes. For an individualized liver-on-a-chip, the target cells must be obtained from patients. Evidence suggests the possibility of transdifferentiating monocytes or fibroblasts into hepatocyte-like cells. Due to their limited cell proliferation capability, these cells must be induced to differentiate into pluripotent stem cells that proliferate indefinitely. Thus, iPS cells provide a scalable assay if needed. Given the low invasiveness of sample collection, blood cells are promising targets for hepatocyte generation. Abbreviations: HSPCs, hematopoietic stem and progenitor cells.

**Figure 4 micromachines-12-00842-f004:**
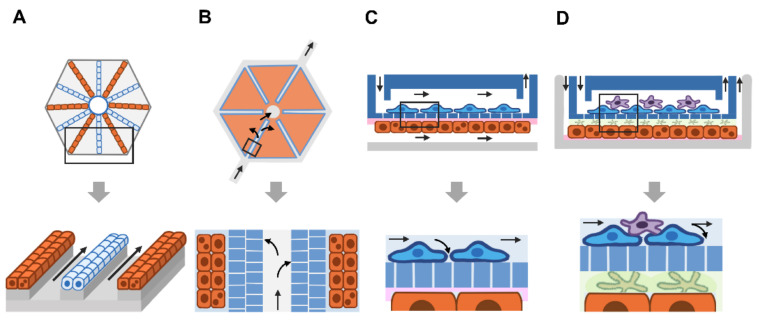
Design of a liver-on-a-chip based on the microarchitecture of the liver lobule. (**A**) The radial pattern of hepatocytes and SECs (light blue-colored ovals) is modeled using electric-based cell seeding. The magnified box shows two lines of hepatocytes and a line of endothelial cells. (**B**) Hexagon-shaped microchip containing six microchambers (red) for cell seeding. The magnified box shows a double porous layer (blue-colored rectangles with white lines) as the fenestrated SECs. Due to their distal and proximal locations to the porous layer of hepatocytes, nutrients and gas diffuse into hepatocytes in a gradient pattern (arrows). (**C**) The two-sided layer allows coculture of distinct cell types. Human primary hepatocytes (orange-colored rectangles) are grown on collagen-coated surfaces (pink), while immortalized bovine aortic endothelial cells are grown in the upper chamber. A porous PET membrane separates the two chambers and allows small molecules to pass through. (**D**) Heterogeneous cell culture in a double microchamber. Human primary hepatocytes (orange-colored rectangles) are grown on the fibronectin-coated surface (pink) of the lower chamber, while the endothelial cell line is grown on the porous PET membrane of the upper chamber. The space between hepatocytes and the porous upper layer is filled with collagen gel (light green) containing hepatic stellate cells (reticular, green-colored shape). The human monocytic cell line U-937 (violet-reticular shapes) was cultured on top of ECs to mimic liver-resident macrophages (Kupffer cells). Arrows indicate the direction of fluid flow.

**Figure 5 micromachines-12-00842-f005:**
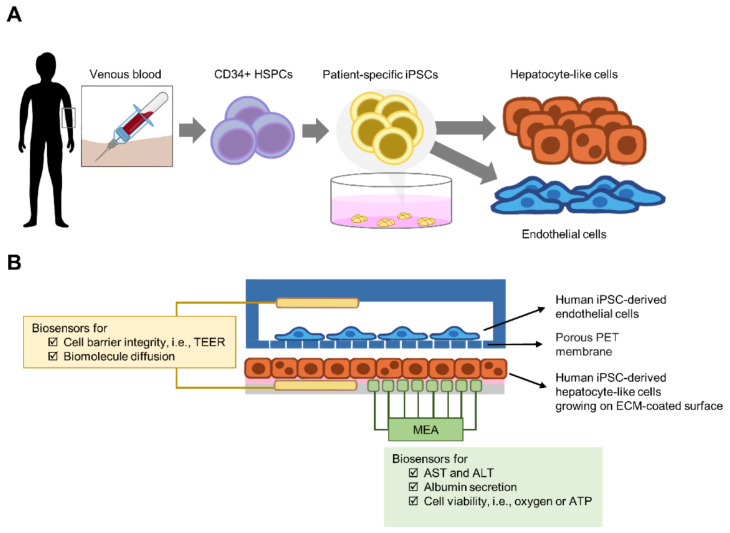
Proposed model for a liver-on-a-chip integrated with a biosensor. (**A**) Collection of CD34+ HSPCs from peripheral venous blood is less invasive than skin biopsy. The use of patient-specific iPS cells is a promising approach to generate individualized hepatocytes and endothelial cells. Moreover, the expandability of iPSCs provides an unlimited supply of patient-specific cells of interest for the liver-on-a-chip. (**B**) Proposed liver-on-chip model. The microarchitecture is similar to that in Figure 3D, except that no monocytic cells or hepatic stellate cells were used. Endothelial cells grew on the porous PET membrane, while human primary hepatocytes grew on the fibronectin-coated surface of the lower chamber. Based on the model proposed by Maoz et al. two different electrodes are placed at distinct positions. The transepithelial electrical resistance (TEER, yellow) electrode was inserted into the upper and lower chambers. Gap formation between endothelial cells causes a decrease in TEER. Thus, sinusoid integrity and other diffusing biomolecules can be examined. Moreover, a multielectrode array (MEA, green) can be inserted in the lower chamber to detect oxygen or other secreted molecules, i.e., AST, ALT or albumin. In addition, the MEA is able to measure morphology, growth and proliferation of cells. Thus, this proposed model is useful for assessing the effects of barrier loss on hepatocyte viability or function. Abbreviations: PET, polyethylene terephthalate; ECM, extracellular matrix.

**Table 1 micromachines-12-00842-t001:** Biosensors available for the assessment of liver functions.

Functions	Recognition Compartment	Surface for Immobilizationand Transduction	Ref.
(1) Cholesterol (2) Bilirubin (3) ALT (4) AST	Enzymes	- Planar, porous silicon chip (working Pt electrode) - Electrochemical transduction (electron or 2e^−^ transfer from H_2_O_2_ to OH on the electrode surface	[88]
Human serum albumin (HSA)	Antibody	- Au electrodes - Glass sensing surface for antibody immobilization	[89]
	Electrochemical sensing	- Covalent bonds between albumin and the Au electrode	[90]
Oxygen	Phosphorescence probe	Microbeads	[91]

Polydimethylsiloxane (PDMS); alanine aminotransferase (ALT); aspartate aminotransferase (AST); human serum albumin (HSA).

**Table 2 micromachines-12-00842-t002:** List of commercial biosensors and their applications in the assessment of cellular barrier integrity.

Commercial Name of TEER Device	Electrodes for the Apical-Basolateral Compartment	Culture Well Format	Applications and References
Epithelial Volt-ohmmeter (EVOM)	A pair of two electrodes (chopstick electrode)	Transwell	BBB using immortalized human brain capillary endothelial cell lines [133]
REMS Auto-Sampler (World Precision Instruments)	Chopstick electrode located on the robotic arm and the REMS electrode interface	Transwell	(1) BBB using murine brain microvascular endothelial cells [134] (2) Gastrointestinal model using human colon adenocarcinoma cell line Caco-2 and endothelial cell line EA.hy926 [135]
EndOhm (World Precision Instruments)	Concentric electrodes on the top and bottom	Single chamber	Renal epithelial barrier using Madin Darby canine kidney (MDCK) strain 1 cells and porcine epithelial kidney cells (LLC-PK1) [137]
ECISA pplied BioPhysics	A gold electrode-containing film	Single chamber	BBB using human brain microvascular endothelial cells [136]

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
