# Peer review of "Progress and Challenges in the Use of a Liver-on-a-Chip for Hepatotropic Infectious Diseases"

_micromachines, 2021, doi:10.3390/mi12070842_

Round 1
Reviewer 1 Report
The authors reviewed the field of liver-on-a-chip for hepatotropic infectious diseases. This review paper might be helpful for potential readers to learn the structures of liver lobules and liver-on-a-chips. Also, the progress and challenges in the field of liver-on-a-chips are well-reviewed.
However, I recommend authors edit the introduction and some paragraphs to explain more clearly.
Major revision
- In the introduction part, it is hard to know what the authors want to tell. Compared to the abstract, the sentences are less related and hard to read. I think the authors should explain more. For example, it is hard to know in this introduction why personalized medicine is important in this field.
- I recommend authors write the main claim in the first sentence of paragraphs. It is hard to anticipate what authors want to claim in many paragraphs before reading all the paragraphs. For example, in line 61, the third sentence is the main claim of the paragraphs. The first and the second sentences should be deleted or written in the middle of paragraphs if the authors think those sentences are necessary.
- The authors should explain more about the necessity of personalized medicine. Although there is a sector named “Necessity of personalized medicine,” the necessity is not written well. Therefore, authors should explain more about personalized medicine and its necessity.
- The authors explained well the severity of hepatotropic infectious diseases in sector 2 (Necessity of personalized medicine for hepatotropic infectious diseases). However, I think it is better to write it in the introduction or at least before sector 1 (Basis of liver microarchitecture for liver-on-a-chip design).
- In the introduction, I think it is better to cite more papers explaining organoids and organ-on-chip (Line 46). Organoids and organ-on-a-chip are huge fields, and they are related a bit to the main claim of this paper, so I recommend authors cite more papers related to this.
- I think it is better to explain more about liver-on-a-chip in sector 4 (line 292). Although the paper's title is progress and challenges in the use of liver-on-a-chip, sector 4 lacks information. Authors can select to explain many things here. For example, authors can explain more about the way to prepare multi-cell culture chip or validation experiments and results of the liver-on-a-chip.
- It is better to draw figure 5 again to emphasize the difference to Fig.3D.
- Several sentences are hard to understand. For example, it is hard to understand the sentence in line 29, “These life-threatening issues raise concerns in developing new tools that must be dealt with.”
Reviewer 2 Report
The manuscript “Progress and challenges in the use of a liver-on-a-chip for hepatotropic infectious diseases” by Kasem Kulkeaw and Worakamol Pengsart review the recent uses of liver-on-a-chip examples of hepatotropic diseases and technologies that integrating biosensor technologies with the liver-on-a-chips. Starting with the fundamentals of liver microarchitecture for liver-on-a-chip design, the authors discussed the need for personalized medicines for hepatotropic infections, liver-on-a-chip, and biosensor assays for liver function. The authors also reviewed the source of hepatocytes for the liver-on-a-chip and addressed the prospects and challenges of the liver-on-a-chip for personalized medicine. The workflow was straightforward, the rationale and the target application are clear; however, there are some concerns to be carefully addressed for being published.
1. There are several grammatical errors throughout the entire manuscript, especially the use of articles (a, an, the) and the use of the comma(ex. a, b, and c). This must be addressed before publication to ensure the quality of the journal.
2. Figure 1.
2-1) It would be more helpful for the readers if Figure 1 is rearranged since the letter c is before the letter b.
2-2) The target body of Figure 1 is quite broad. No sentences specifically referencing figure 1 a, b, or c in the manuscript.
3. Formats should keep consistency as journals standard. Some components have bigger font than others. For instance:
line 221 Plasmodium vs 222 Plasmodium
line 227 Anopheles
line 297 liver-sinusoid-on-a-chip
4. Overall, the texts in the figures are hard to be seen. Please use the journal-recommended fonts with sizes.
5.Compared to the importance emphasized in the introduction, the case of “Liver-on-a-chip for hepatotropic infectious diseases” was not emphasized and addressed less significantly.
6. The examples of a biosensor in the section "Biosensor assays for liver function" are quite old. This will especially diminish the interest of readers of "Micromachines".
6. In conclusion, the authors mention TEER and MEA as promising technologies for Liver-on-a-chip technology. However, only brief possibilities of these technologies are presented, few examples are introduced, and special merits applicable to Liver-on-a-chip technology are not discussed in depth. It would be great if the manuscript could take a more in-depth approach to the technologies so that readers of "Micromachines" can enjoy the manuscript.
Round 2
Reviewer 1 Report
The authors edited the manuscript well according to my comments and suggestions. The introduction is more readable and understandable. I hope this manuscript can help potential readers to understand the field of liver-on-a-chips.
Reviewer 2 Report
The authors well revised the manuscript. Now it looks suitable for publication.